# Tobacco Whack-A-Mole: A Consumption Taxonomy of Cigar & Other Combustible Tobacco Products among a Nationally Representative Sample of Young Adults

**DOI:** 10.3390/ijerph192215248

**Published:** 2022-11-18

**Authors:** Kymberle Landrum Sterling, Katherine Masyn, Stephanie Pike Moore, Craig S. Fryer, Erika Trapl, Ce Shang, Douglas Gunzler

**Affiliations:** 1Department of Health Promotion & Behavioral Sciences, School of Public Health, University of Texas Health Sciences Center, Dallas, TX 75207, USA; 2Department of Epidemiology and Biostatistics, School of Public Health, Georgia State University, Atlanta, GA 30302, USA; 3Department of Population and Quantitative Health Sciences, Case Western Reserve University, Cleveland, OH 44106, USA; 4Department of Behavioral and Community Health, School of Public Health, University of Maryland, College Park, MD 20742, USA; 5Department of Internal Medicine, Wexner Medical Center, Ohio State University, Columbus, OH 43210, USA; 6Department of Medicine, The MetroHealth System, Case Western Reserve University, Cleveland, OH 44106, USA

**Keywords:** cigarillo, little filtered cigar, cigarette, large cigar, tobacco use, young adults

## Abstract

Introduction: Little filtered cigars and cigarillos (LCCs) are consumed infrequently, co-administered with marijuana, and concurrently used with other tobacco products. Reliance on the past 30-day use estimate, a marker of tobacco user status, may underestimate the dynamic nature of intermittent LCC and other tobacco product use. We developed a framework to capture the intermittent nature of exclusive LCC use and dual/poly use with cigarettes and large cigars using broader timing of last product use categories and product use modality (e.g., with marijuana). Methods: Data come from the baseline C’RILLOS study, a U.S. nationally representative sample of young adults aged 18–34 (*n* = 1063) collected in October 2019. We developed a consumption taxonomy framework that accounted for respondents’ modality of LCC use (i.e., use with tobacco, LCC-T, or use with marijuana as blunts, LCC-B), the exclusive use of LCCs and other tobacco products (i.e., cigarettes, and large cigars) or their co-use and the timing of last product use (i.e., ever and past 30 days, past 3 months, past 6 months, greater than 6 months). Results: Seventy-five percent of our sample reported ever use of any combustible tobacco product, including LCCs. The most common ever use pattern was poly use of LCC-T + LCC-B + cigarettes (16%). Our consumption taxonomy framework demonstrated the fluid nature of combustible tobacco product use among LCC users. For instance, among past 30-day cigarette users, 48% reported using LCC-T, 39% reported using LCC-B, and 32% reported using large cigars in the past 3 months or more. Discussion: The tobacco use field currently classifies ‘tobacco users’ based on the product they smoked in the past 30 days. Any tobacco product use beyond the past 30-day period is considered ‘discontinued use’ and not the focus of intervention or tobacco regulatory science decisions. We documented the substantial proportion of young adult LCC, cigarette, and large cigar users who either exclusively or dual/poly used these combustible products in recent (e.g., past 3 months) periods. To prevent underestimation of use, surveillance measures should assess the use modality, timing of last product use, and exclusive/multiple product use to more accurately identify the smoking status of young adult LCC users.

## 1. Introduction

After remaining consistent for a decade [1], a significant uptake in filtered little cigars and cigarillos (LCCs) by young adults occurred between 2008 and 2015 [2]. Data from Wave 3 of the Population Assessment of Tobacco Use or Health (PATH) study indicate that 14.0% of young adults aged 18–24 reported past 30-day use of cigar products. Of those past 30-day cigar product users, 44.8% reported using cigarillos, and 4.1% reported using little filtered cigars. Roughly 35% of the past 30-day cigar users reported using two or more cigar products [3]. Evidence indicates that cigars may be used intermittently or occasionally across all age groups [4,5,6], and additional cigar smoking surveillance is needed to understand variations in use patterns. The past 30-day use measure, or product use “every day” or “someday” within the past 30 days, are two measures that commonly operationalize current use and characterize intermittent or occasional cigar use [7]. However, the current cigar use assessment is based on cigarette smoking patterns. Reliance on cigarette use measures to assess cigar smoking may underestimate patterns of use [8], especially for a class of combustible tobacco products that users smoke inherently differently than cigarettes.

Cigarillo smokers have reported consuming only part of the product in a smoking session. Instead, these smokers use a portion of it, save the remaining amount, and re-light it for use at a later time [9,10]. Situational factors that characterize LCC use differently from cigarette use, such as primarily using the products in social settings and sharing them with others [11,12] may contribute to the intermittent nature of the product’s use.

Cigarillos are often co-administrated with marijuana and smoked as “blunts” [10,13,14]. To make a blunt, the cigarillo is cut open, and all or some of its tobacco is replaced with marijuana. Data from Wave 3 of the PATH study indicate that 52.1% of young adults had ever used marijuana in 2015–2016. Cigars were the most common product used for marijuana consumption (i.e., blunt smoking) among young adult marijuana ever users; 74% reported ever using a blunt. Because of the high prevalence of ever blunt use among young adults and the evolving patterns of marijuana and tobacco use among this group, researchers have called for studies and measures that assess blunt use recency [15].

Some LCC users (i.e., with its tobacco or as blunts) are likely to engage in dual or poly-tobacco product use. Dual- and poly-tobacco product use, or the use of two or more products, is increasing [16,17,18] and is more prevalent among younger than older adults [3]. Data from Wave 4 of the PATH study indicate that 33% of tobacco-using adults, ages 18 and older, were past 30-day multiple tobacco product users. The dual use of cigarettes and cigarillos was the most commonly reported combustible multiple tobacco use pattern among those adults, with 32.3% dual use of these products [19]. Moreover, the financial status of some price-sensitive LCC smokers may also influence episodic use among dual- or multi-tobacco product users. Recently, Ganz and colleagues noted that dual cigarette and cigarillo users might purchase cigarillos that are often cheaper than a pack of cigarettes “to keep costs down” [20].

These studies indicate that LCCs may be used sporadically over extended periods for some users (e.g., dual- or multi-tobacco product users). Instead of being characterized as a discontinued behavior or quitting [3], LCC use that occurs beyond past 30-day or every day/someday use may be better categorized as use over an extended period. The scenarios highlight the need for measures that capture use beyond the past 30-day or every day/someday use.

The Food and Drug Administration (FDA) proposes to ban characterizing flavors (e.g., fruit, candy, etc.) in cigars [21], emphasizing the need to assess the impact of the product standard on the use patterns of the legally remaining LCCs products (e.g., tobacco-flavored) on the market. Once the flavor ban is implemented, relying on estimates of past 30-day use, which often guide regulatory policy actions and conservatively capture tobacco use behaviors, may under-estimate the dynamic and complex pattern of LCC use and its intermittent co-use with other tobacco products. This paper introduces a broader methodological and measurement framework that offers a different perspective on how LCC smoking can be operationalized beyond the past 30-day use. To capture the variability in young adults’ LCC use patterns more accurately, the framework proposes measures that comprehensively assess cigar users’ “time since last LCC smoking episode”, their LCC product use modality (i.e., with tobacco or marijuana), and co-use of other combustible tobacco products. The framework provides practical ways to assess patterns of LCCs use and may provide a more comprehensive impact evaluation of tobacco regulatory policies.

## 2. Material and Methods

### 2.1. Study Overview

Study data were collected as a part of The C’RILLOS Project, a mixed methods research study that examines the impact of cigar product packaging features on young adults’ receptivity, risk perceptions, and future use of LCC smoking behaviors. Data presented in this manuscript come from the baseline survey, collected in October 2019, of our online longitudinal survey. The survey sought to assess LCC smoking behavior, advertising exposure, and risk perceptions about use among a nationally representative sample of 18 to 34-year-olds in the United States. The study was approved by the University of Texas Health Science Center’s Institutional Review Board.

### 2.2. Data Source

The C’RILLOS Project cohort sample comprises 1123 young adults aged 18 to 34 residing in the United States (U.S.). The sample was selected from NORC’s AmeriSpeak Panel. Funded and operated by NORC at the University of Chicago, AmeriSpeak^®^ is a probability-based panel designed to be representative of the U.S. household population. Randomly selected U.S. households were sampled using area probability and address-based sampling, with a known, non-zero probability of selection from the NORC National Sample Frame. These sampled households were contacted by U.S. mail, telephone, and field interviewers (face to face). The panel provides sample coverage of approximately 97% of the U.S. household population. Those excluded from the sample include people with P.O. Box only addresses, some not listed in the USPS Delivery Sequence File, and some newly constructed dwellings. While most AmeriSpeak households participate in surveys via the web, non-internet households participate in AmeriSpeak surveys by telephone. Households without conventional internet access but having web access via smartphones were allowed to join in AmeriSpeak surveys online.

### 2.3. Sample Selection

The sample for The C’RILLOS Project was selected from the AmeriSpeak Panel using sampling strata based on age, race, Hispanic ethnicity, education, and gender. We oversampled for Non-Hispanic African Americans as they are at heightened risk for LCC use [10]. At baseline, 4794 NORC panelists were invited to complete the survey. Of those, 1234 completed screening interviews, resulting in a screener completion rate of 25.7%. Of those who completed the screener, 1195 (96.8%) were eligible for the survey. Of those eligible, 1123 (94.0%) completed the baseline survey; 60 were excluded due to inconsistent responses, yielding a total analytic sample of 1063.

### 2.4. Survey Procedure

Before the survey launched, its wording and question structure were pretested among a small sample of panelists. Revisions were made as necessary in response to the pretest results. Next, participants were screened for study eligibility. The screening and main survey stages of data collection were conducted during a single survey session for the respondents. Study panelists who were English or Spanish-speaking young adults between 18–34 years were eligible to participate in the survey. We selected this age range based on prior studies that indicated this group had a heightened risk of LCC use [10]. Young adults with a history of LCC use were eligible to participate in the study. Young adults who had never used LCCs were also eligible to participate in the study. We included LCC non-users in our sample to document their onset of LCC use in subsequent survey waves. Screened and eligible panelists participated in the second stage, which was the main study. The online survey was offered in English and Spanish and was fielded for approximately three weeks. The median length of the survey was 18–20 min.

### 2.5. Study Measures

#### Sociodemographic Characteristics

Sociodemographic characteristics examined included age, gender (male, female), combined race and ethnicity (non-Hispanic White, non-Hispanic Black, non-Hispanic Other, Hispanic, non-Hispanic Multiracial, non-Hispanic Asian), sexual orientation (heterosexual/straight, homosexual/gay or lesbian, bisexual, self-identify, or do not know/not sure), educational attainment (less than high school, high school graduate or equivalent, vocational/tech school/some college/Associate’s degree, Bachelor’s degree or more), employment status (working as a paid employee or self-employed, not working on temporary layoff from a job or looking for work, or not working because retired, disabled or other), income, and geographic region.

### 2.6. Cigar Use Modality and Combustible Tobacco Product Use

LCCs, cigarettes, and large cigars were the primary combustible tobacco products of interest. Consistent with prior studies documenting the importance of including brand-specific names to aid recall [22,23], respondents were provided with the brand names of popular LCC and large cigar products and shown images to assist with product identification. For LCCs, respondents were asked about two modalities of use: “as sold” (i.e., used with its tobacco, LCC-T) and as blunts” (i.e., used with marijuana, LCC-B). Respondents who said yes to “Have you ever smoked a little cigar or cigarillo without marijuana (weed, pot, loud, etc.) inside it?” were defined as ever LCC-T users. Respondents who said yes to “Have you ever smoked a little cigar or cigarillo with marijuana (weed, pot, loud, etc.) inside it?” were defined as ever LCC-B users. Respondents who said yes to ever smoking all or part of a large cigar (such as Cohiba, Macanudo, and Arturo Fuente brands) at least once in their lifetime were defined as ever large cigar users. Our large cigar measure used premium brand names, such as Cohiba, Macanudo, and Arturo Fuente, to assist respondents with product category identification. Although premium cigar brands were used as examples, the use of non-premium large cigars also may have been captured. Respondents who said yes to ever trying cigarette smoking, even one or two puffs, were defined as ever cigarette users.

### 2.7. Number of Combustible Tobacco Products Used

Respondents who said “no” to all combustible tobacco product use questions were defined as never combustible tobacco users. Respondents who indicated they had ever used only one product (LCC-T, LCC-B, large cigars, or cigarettes) were defined as ever exclusive users of those products. Respondents who indicated they had ever used two combustible tobacco products were defined as ever dual users. Those who reported smoking more than two combustible products were defined as ever polytobacco users.

### 2.8. Timing of Last Combustible Tobacco Product Use

For each product that a respondent indicated that they had ever used, they were asked to report the last time they had used the product and were provided with the options: within the past 30 days, within the past 3 months, within the past 6 months, within the past year, within the past 5 years, and longer than 5 years ago.

### 2.9. Data Analysis

Data management and descriptive statistics, including weighted frequencies, percentages, means, and standard deviations, were done using SAS version 9.4 (SAS Institute Inc., Cary, NC, USA). The weights were constructed to U.S. population benchmarks aged 18–34 among Hispanic, Non-Hispanic African American, and Non-Hispanic other races. Within each of these race/ethnicity groups, the sample was adjusted using the Current Population Survey 2019 (March Supplement) by age, gender, education, and census region. In addition to the above adjustment, weights for the study were also adjusted via a raking ratio method to the U.S. population aged 18–34 on race/ethnicity groups.

## 3. Results

### 3.1. Participant Sociodemographic Characteristics

Our sample was 26 years of age; half were male, 45% were non-White, and 84% identified as heterosexual/straight. Around 71% of respondents reported having some college education or more, 73% were employed, and 38% reported being from the southern region of the U.S. (Table 1).

### 3.2. Ever Combustible Tobacco Product Use

In examining patterns of combustible product use across our sample, 15 unique combinations of ever use were identified and summarized across single (or exclusive), double (or dual), triple, and quadruple (or poly) product ever use (Table 2) (Never users are not included in the number of combinations.). One quarter (25%) had never used any of the four combustible tobacco products, 57% had ever used cigarettes, 47% had ever used LCC-T, 45% had ever used LCC-B, and 24% had ever used large cigars. Exclusive ever use of any one of the products was reported among 20% of the sample. Among the exclusive ever users, half reported exclusive ever use of cigarettes. The multiple product ever use (i.e., use of two or more products) was reported for 56% of the sample. The most common ever use pattern was using LCC-T, LCC-B, and cigarettes (16%). Ever users were twice as likely to be multiple product users (56%) compared to exclusive product users (25%).

### 3.3. Timing of Last Combustible Tobacco Product Use

Figure 1 presents data on the last time respondents reported using cigarettes, LCC-T, LCC-B, and large cigars. While a larger proportion of respondents reported last using these combustible tobacco products within the past year or longer, unique frequency distributions were observed for cigarettes, LCC-T, LCC-B, and large cigars within the past 30-day to past 6-month period. For cigarettes, nearly a third of respondents (28%) reported last using cigarettes within the past 30 days, and 10% reported cigarette use across the past 3 to 6 month period. However, LCC-T and LCC-B smokers exhibited more variation in their time since last use distribution than cigarette smokers; their time since last product use was evenly distributed across the past 30 days to the past 6 months. Among LCC-T smokers, 14% reported last smoking them within the past 30 days, while 18% reported last use within the past 3–6 months. Nearly one-third (29%) of LCC-B smokers reported last using them within the past 30 days, while 21% reported last use in the past 3–6 months. Among large cigar users, 7% reported past 30-day use of large cigars, and 27% reported use in the past 3–6 months.

### 3.4. LCC Use Heterogeneity: Modality and Timing of Last Use

Figure 2 illustrates the complexity of LCC use by modality and the last time respondents’ smoked the product, which includes using up to the past 6 months. We examined use up to the past 6 months because, as noted in Figure 1, it reflects our respondents’ commonly reported LCC use periods. A total of 15 mutually exclusive types of LCC user groups were found, each representing respondents with unique behavioral profiles (A never user category was found, but are not included in the total number of categories). For example, among individuals who have used LCC by any modality in the past 30 days (weighted *n* = 195), 15% (*n* = 30) exclusively used LCC-T in the past 30 days and were exclusive LCC-T ever users (see 2.1); 31% (*n* = 60) exclusively used LCC-B in the past 30 days and were exclusive LCC-B ever users (see 2.2); 15% (*n* = 29) exclusively used LCC-T in the past 30 days, but had used LCC-B more than 30 days ago (see 2.3); 32% (*n* = 63) exclusively used LCC-B in the past 30 days, but had used LCC-T more than 30 days ago (see 2.4); and 7% dual used LCC-T and LCC-B (*n* = 14) within the past 30 days (see 2.5).

### 3.5. Conceptual Framework of Consumption Taxonomy

We developed a conceptual framework to account for LCC modality (i.e., LCC-T, LCC-B), timing of product use, and the number of combustible products smoked. First, we examined the number of combustible tobacco products the respondents smoked: none, one (i.e., exclusive use), two (i.e., dual use), and more than two products (i.e., poly use). Next, we assessed the respondents’ reported number of products smoked, their modality of LCC used, and their reported concurrent use with other combustible tobacco products (e.g., dual-use with LCC-B and large cigars or poly use with 3 or more products). Finally, respondents’ number of products smoked, their LCC modality, and concurrent use were examined across each reported use period (e.g., exclusive LCC-B in the past 30 days but poly-use with 3 or more products in the past 3 months).

#### 3.5.1. Consumption Taxonomy: Exclusive Product Use across each Reported Use Period

Overall, exclusive use and dual/poly product use of cigarettes, LCC-T, LCC-B, and large cigars were found in each period. Compared to exclusive users of cigarettes and LCC-T, a higher proportion of LCC-B and large cigar users reported exclusive product use in the past 30 days. A similar pattern was found in the past 3 months. However, a higher proportion of cigarette users, compared to the other combustible users, reported exclusive use during the past 6 months or more (Table 3).

#### 3.5.2. Consumption Taxonomy: Dual/Poly during Past 30 Days before Survey

Sixty-nine percent of LCC-T, 48% of LCC-B, and roughly 75% of large cigar users reported dual- or poly-use of any combustible tobacco product within the past 30 days before the survey. When examining dual use with cigarettes, 55% of past 30-day LCC-T users, 45% of past 30-day LCC-B users, and 38% of past 30-day large cigar users reported co-using (or dual use) with cigarettes in the past 30 days. A third of past 30-day LCC-T and just over a third of past 30-day LCC-B and cigarette users reported dual or poly use of any combustible product more than 30 days ago; only 10% of past 30-day large cigar users reported this. Notably, 48% of past 30-day cigarette users reported use of LCC-T more than 30 days ago. Similarly, 46% of past 30-day LCC-B and a quarter of past 30-day large cigar users reported using LCC-T more than 30 days ago.

#### 3.5.3. Consumption Taxonomy: Dual/Poly during the Past 3 Months before Survey

The prevalence of dual- or poly-combustible product use in the past three months before the survey was low among LCC-T, LCC-B, and large cigar users. Nearly 10% of past 3-month LCC-T and LCC-B users dual- or poly-used any combustible product use in the past three months; even fewer (7%) of past 3 months large cigar users dual- or poly-used any combustible product use in the past three-month period. A larger proportion of past 3-month cigarette users (18%) dual- or poly-used other combustible tobacco products in the past 3 months. However, those with a history of past 3-month use also reported more recent past 30-day use of any combustible tobacco product. Over half of past 3-month cigarette users, 41% of past 3-month LCC-T, and 37% of past 3-month large cigar users reported dual-or poly-use of another combustible tobacco product in the past 30 days.

#### 3.5.4. Consumption Taxonomy: Dual/Poly Use during the Past 6 Months before Survey

Compared to past 6-month LCC-B (12%) and large cigar (18%) users, past 6-month cigarette (21%) and LCC-T (22%) users had the highest prevalence of dual- or poly-use of any combustible product during the same time frame. Past 6-month LCC-T (57%), LCC-B (48%), and large cigar users (61%) had a greater likelihood of dual- or poly-using another product more recently (less than 6 months) than past 6-month cigarette users (26%).

#### 3.5.5. Consumption Taxonomy: Greater Than 6 Months before the Survey

Among respondents who reported smoking more than six months before the survey, nearly half or more of cigarette, LCC-T, LCC-B, and large cigar users reported dual- or poly-using another combustible tobacco product in the same period. Large cigar (43%), LCC-T (40%), and LCC-B (39%) users were more likely to have dual- or poly-used another product more recently (in the past 6 months or less) compared to cigarette users (28%).

## 4. Discussion

We report data on the patterns of LCC use that emerge when considering young adults’ product modality (i.e., LCC-T, LCC-B), dual/poly-use of LCCs and other combustible products, and time since last smoked to elucidate the dynamic and often fluid smoking behavior of cigar product users. Nearly half of the C’RILLOS young adults have a history of LCC-T and LCC-B use, but some reported concurrent dual/poly-using with cigarettes and large cigars. LCC-T and LCC-B smokers also engaged in “on again” and “off again” use of cigarettes and large cigars over an extended period, including within the past 30 days, past 3–6 months, or longer than 6 months before the survey. To our knowledge, this is the first study to report the dynamic smoking behavior of young adults with a history of LCC-T and LCC-B use. As such, we refer to this fluid smoking behavior as the “Tobacco Whack-A-Mole”: the game where the ‘mole’ (young adult user) pops in and out of a different ‘hole’ (product).

Young adult ever smokers were more than twice as likely to be multiple product users (over 50%) than exclusive users (20%) of those products. The most common multiple product ever use pattern was the concurrent use of LCC-T, LCC-B, and cigarettes. While prior studies have found that the most commonly used multiple combustible tobacco product use combination among U.S. adults was cigarettes and cigars [16], our study contributes to the field by specifying the modality of cigar use.

An important but unexpected finding was the reported variation in the time our respondents reported last using LCCs, cigarettes, and large cigars. As noted in Figure 2, nearly a third of cigarette and LCC-B users’ reported smoking these products within the past 30 days before the survey. However, LCC-T users’ reported smoking the products beyond the past 30-day period. Eighteen percent of LCC-T use was reported within the past 3- to the 6-month time frame before the survey. A similar phenomenon was found for large cigars, with nearly a third reporting use within the past 3- to 6 months before the survey. While other studies have found that cigar smokers report using the products less frequently than cigarettes [5], our study sheds light on when young adults report smoking cigars and other combustible products. Hyland et al. (2020) describe three mutually exclusive tobacco use states—never, current, and former—through which users may continuously transition over time [24]. We cannot capture smoking transitions with cross-sectional data. However, our findings indicate that young adults, including cigar dual/poly-combustible product smokers, vary in their timing of product use. Future studies should examine this phenomenon among other cigar-using young adult samples. A practical approach to conducting future studies like these is using the broader, extended “time since last use” item. Along with understanding why young adult cigar smokers use these and other combustible products at varying times, future studies are also needed to understand if smoking time variation is an important precursor to transitions in tobacco use states among young adults.

Current or past 30-day use is often used as an indicator of smoking behavior status, and its rates can inform prevention or cessation intervention development and influence tobacco regulatory policy decisions. Typically use beyond the past 30 days is considered discontinued use. If the tobacco control field continues to utilize the past 30-day use as a “status quo” marker of prevalence and considers use beyond the past 30 days an indicator of quitting, intervention and policy strategies may not be inclusive of infrequent users who may still be nicotine-dependent. Although nicotine is metabolized and removed from the body 72 h after one quits smoking, cotinine may be detectable in the body for up to 3 weeks after the last exposure. Moreover, human brain chemistry does not return to normal until at least 3 months after nicotine use [25]. Thus, although cigar use may be infrequent or occur over extended periods, its abuse liability remains problematic, even beyond the past 30-day period, providing some justification for the importance of assessing user status beyond the past 30 days.

We developed a consumption taxonomy framework that accounts for how our young adult sample used LCCs (e.g., LCC-B), if they used the products exclusively (e.g., LCC-B only) or co-used them with other products (e.g., LCC-B and cigarette use), and when they used LCCs and other products of interest (e.g., smoked cigarette use in the past 30 days but used LCC-B in the past 3 months). The use of the consumption taxonomy framework highlighted the fluid nature of dual- and multi-product use among some LCC users. For instance, while over half of our cigarette, LCC-T, and LCC-B ever users reported concurrently using any combustible tobacco product in the past 30 days, over a third of these smokers also reported dual- or poly-using another combustible product within the past three months or longer. The framework’s utility is its ability to conceptualize the dynamic nature of young adults’ episodic smoking behavior, demonstrating how their smoking patterns are fluid and multifaceted.

The consumption taxonomy framework also shows how the sole reliance on the past 30-day status marker may lead to misclassifying multiple tobacco product users as exclusive product users, especially cigar smokers. Longitudinal data examining the PATH study’s exclusive and poly-tobacco cigar use demonstrates this in its measurement. Individuals who had previously used a cigar product in the past 30 days, but did not in subsequent waves, were considered discontinued users. Additionally, their subsequent reported use in the past 30 days was considered reuptake [3]. However, what has previously been classified as “discontinued” use may reflect a differential use pattern. Differential patterns of use can be explained by personal-level factors (i.e., nicotine dependence), person-product level factors (i.e., perceived harm), situational factors (i.e., product availability, social group norms), as well as the dynamic complementarity of the products themselves [19]. Our consumption taxonomy framework highlights the shifting consumption and dynamic complementarity of LCC-T, LCC-B, large cigar, and cigarette product use among young adults.

### Implications for Tobacco Regulatory Science

Product standards enacted on a single product may have unintended effects on other smoking behavior because of dual/poly-tobacco product use. For instance, the use of one product may shift based on product standards enacted on another; this has been demonstrated for policies that restrict the use of flavor in tobacco products [26,27] and those that seek to increase product price through excise taxes [28]. The measurement of tobacco use behavior directly impacts policies designed to improve public health. Our findings suggest that the category of cigars, which includes cigarillos used with tobacco, cigarillos used with marijuana as blunts, and large cigars, including non-premium and premium brands, should be measured through an extended time frame because their usage patterns differ from other tobacco products (i.e., cigarettes). The suggested change in measuring cigar use may impact the evaluation of existing or future tobacco regulatory policy decisions (e.g., advertising restrictions)—thus assessing the impact of the policy on usage rates over an extended period (i.e., 3 months vs. past 30 days). Broadly, the field should consider expanding its surveillance measures to capture respondents’ “timing of last product use”, a measure that would incorporate and go beyond past 30-day use, to better capture the nuanced and shifting nature of use behaviors in young adults. Doing so will improve the evaluation of current and future tobacco control policies.

## 5. Limitations

It is beyond the scope of our study to say that extended use of LCCs is predictive of continued product use. However, we acknowledge that our findings highlight a need to understand the association between time since last use and continued product use. Additionally, we call for future studies to examine the association between these variables and situational factors, such as price and social use of the products. Additionally, data were collected before the COVID-19 pandemic, which has been shown to disrupt patterns in tobacco use [29,30,31]. Furthermore, these data are cross-sectional. Longitudinal research is needed to understand how individuals move in and out of different modalities and recency patterns.

## 6. Conclusions

Using a consumption taxonomy framework, our study is the first to describe the time-varying modalities of LCC-T and LCC-B with the co-occurrence of cigarette and large cigar tobacco beyond ever and past 30-day use. Our framework highlights the complexity of LCC and other combustible smoking behavior among young adults. The past 30-day use measure of LCCs and other combustible products may not be a comprehensive assessment of product use. Young adult smoking patterns are shifting from exclusive product use and more likely exist within a fluid gradient of multiple product use than previously described. The dynamic pattern of tobacco product use has substantial implications for measuring smoking behavior, developing cessation efforts, and anticipating the behavioral response of tobacco regulatory policy implementation.

## Figures and Tables

**Figure 1 ijerph-19-15248-f001:**
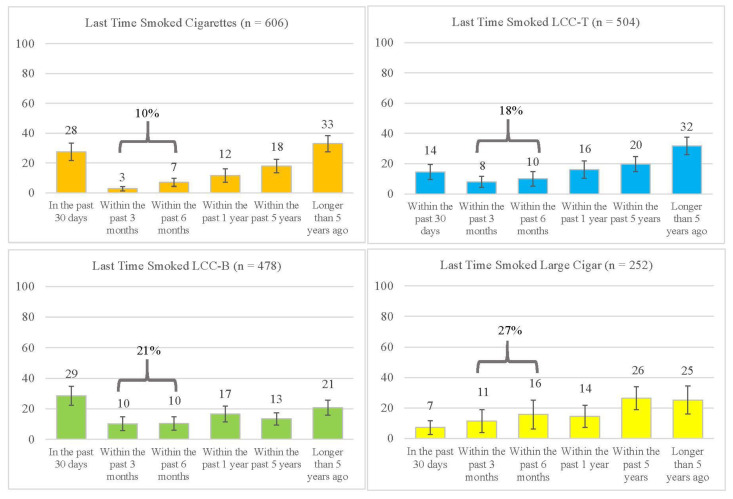
Time of Last Use of Cigarettes, LCC-Tobacco, LCC-Blunt, and Large Cigars among the C’RILLOS Sample (*n* = 1063).

**Figure 2 ijerph-19-15248-f002:**
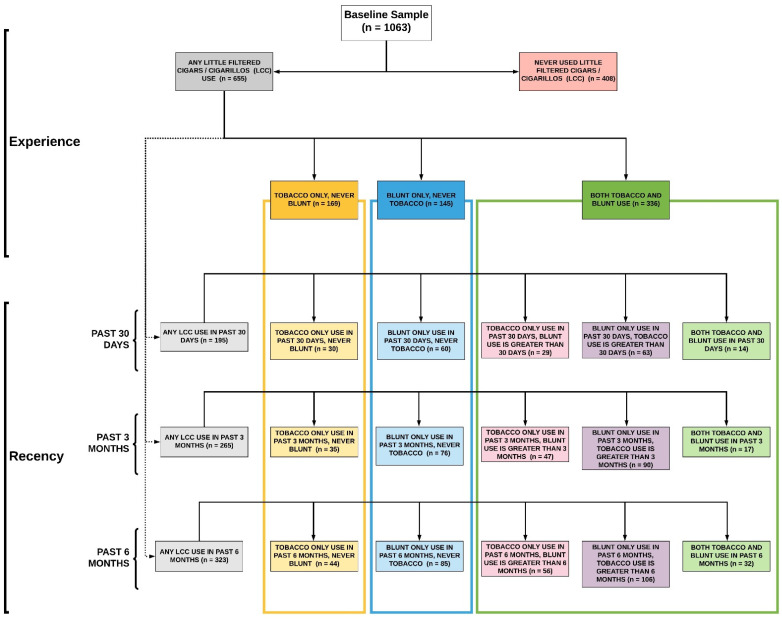
LCC Use Heterogeneity: Modality and Timing of Last Use.

**Table 1 ijerph-19-15248-t001:** Young Adult Sociodemographic Factors by Combustible Tobacco Product Use, Weighted to U.S.-Representative Population.

	Full Sample	Cigarettes	LCC-T	LCC-B	Large Cigars
	*n* = 1063	wt *n* = 606	wt *n* = 505	wt *n*= 481	wt *n* = 252
**Age**, mean	26	27	27	27	27
**Gender**, %					
Male	50	51	54	51	73
Female	50	49	46	49	27
**Race/Ethnicity**, %					
Non-Hispanic White	55	58	55	52	58
Non-Hispanic Black	14	12	12	15	13
Hispanic	22	23	25	25	22
NH Asian	7	4	4	4	4
Non-Hispanic Other/Multiracial	4	3	3	4	3
**Sexual Orientation**,%					
Heterosexual/Straight	84	83	85	84	84
Homosexual/Gay or Lesbian	4	5	3	3	8
Bisexual	8	10	9	9	6
Other	1	1	0	1	0
Do not know/not sure	3	2	2	2	1
**Education**, %					
No HS Diploma	10	11	12	14	16
HS grad or equivalent	28	28	29	28	29
Some college	32	34	32	37	22
BA or above	39	28	27	21	33
**Employment status**, %					
Working	73	73	69	67	78
Not Working: Laid off/looking	14	12	16	19	8
Not Working: Disabled or other	14	14	15	15	14
**Income**, %					
Less than 10 k	6	6	7	10	5
10 k to 25 k	16	16	17	19	16
25 k to 50 k	26	26	27	27	16
50 k to 75 k	24	26	23	19	25
75 k+	27	25	26	25	38
**Region**, %					
Northeast	17	20	20	18	18
Midwest	20	22	21	18	21
South	38	36	35	37	37
West	25	22	25	27	24

**Table 2 ijerph-19-15248-t002:** Ever Poly-Tobacco Product Use Young Adults (*n* = 1063), Weighted to U.S.-Representative Population.

	Pattern	Ever Cigarette	Ever LCC-T	Ever LCC-B	Ever Large Cigar	Young Adult Population (%)
**Never**	**1**					25
**Exclusive/** **Single Product Ever Use**	**2**	**+**				10
	**3**		**+**			3
	**4**			**+**		6
	**5**				**+**	1
**Dual Product Ever Use**	**6**	**+**	**+**			7
	**7**	**+**		**+**		5
	**8**	**+**			**+**	3
	**9**		**+**	**+**		5
	**10**		**+**		**+**	1
	**11**			**+**	**+**	1
**Triple Product Ever Use**	**12**		**+**	**+**	**+**	2
	**13**	**+**	**+**	**+**		16
	**14**	**+**		**+**	**+**	2
	**15**	**+**	**+**		**+**	5
**Quadruple Product** **Ever Use**	**16**	**+**	**+**	**+**	**+**	9
	**Overall**	**57%**	**47%**	**45%**	**24%**	

Color legend: Gray = exclusive/single product ever use. Green = dual product ever use. Blue = triple product ever use. Orange = quadruple product ever use.

**Table 3 ijerph-19-15248-t003:** Consumption Taxonomy of Exclusive, Dual, and Poly Combustible Tobacco Use Among Young Adults, Weighted to U.S.-Representative Population.

	Cigarettes *n* = 606	LCC-T *n* = 504	LCC-B *n* = 478	Large Cigars *n* = 252
**PAST 30 days**				
**Exclusive Use**	5%	2%	18%	16%
**Dual Use, As Recent** **(Within Past 30 days)**				
Cigarette	-	55%	45%	38%
LCC-T	24%	-	10%	38%
LCC-B	37%	19%	-	23%
Large Cigar	4%	9%	3%	-
Dual/Poly Any Combustible Product Use	57%	69%	48%	74%
**Dual Use, Less Recent** **(Greater than Past 30 days)**				
Cigarette	-	35%	26%	24%
LCC-T	48%	-	46%	25%
LCC-B	39%	39%	-	37%
Large Cigar	32%	29%	22%	-
Dual/Poly Any Combustible Product Use	38%	30%	34%	10%
**Past 30-Day Total, n**	167	73	136	18
**PAST 3 months**				
**Exclusive Use**	3%	5%	19%	8%
**Dual Use, More Recent** **(Less than Past 3 months)**				
Cigarette	-	25%	16%	0%
LCC-T	26%	-	4%	21%
LCC-B	37%	35%	-	15%
Large Cigar	0%	3%	0%	-
Dual/Poly Any Combustible Product Use	52%	41%	17%	37%
**Dual Use, As Recent** **(Within Past 3 months)**				
Cigarette	-	11%	2%	5%
LCC-T	26%	-	6%	7%
LCC-B	6%	7%	-	5%
Large Cigar	9%	5%	3%	-
Dual/Poly Any Combustible Product Use	18%	9%	8%	7%
**Dual Use, Less Recent** **(Greater than Past 3 months)**				
Cigarette	-	36%	49%	60%
LCC-T	32%	-	56%	53%
LCC-B	29%	47%	-	37%
Large Cigar	37%	39%	32%	-
Dual/Poly Any Combustible Product Use	28%	44%	56%	48%
**Past 3 months Total, n**	17	40	49	29
**PAST 6 months**	**Cigarettes**	**LCC-T**	**LCC-B**	**Large Cigars**
**Exclusive Use**	19%	10%	6%	0%
**Dual Use, More Recent** **(Less than Past 6 months)**				
Cigarette	-	7%	15%	39%
LCC-T	5%	-	21%	24%
LCC-B	14%	35%	-	33%
Large Cigar	13%	22%	21%	-
Dual/Poly Any Combustible Product Use	26%	57%	48%	61%
**Dual Use, As Recent** **(Within Past 6 months)**				
Cigarette	-	22%	20%	9%
LCC-T	26%	-	30%	14%
LCC-B	23%	30%	-	4%
Large Cigar	8%	11%	3%	-
Dual/Poly Any Combustible Product Use	21%	22%	12%	18%
**Dual Use, Less Recent** **(Greater than Past 6 months)**				
Cigarette	-	37%	29%	26%
LCC-T	40%	-	32%	15%
LCC-B	30%	13%	-	18%
Large Cigar	13%	8%	18%	-
Dual/Poly Any Combustible Product Use	34%	11%	35%	20%
**Past 6 months Total, n**	43	50	50	40
**6 months OR MORE**				
**Exclusive Use**	25%	6%	11%	1%
**Dual Use More Recent** **(Within Past 6 months)**				
Cigarette	-	26%	28%	29%
LCC-T	14%	-	17%	15%
LCC-B	17%	22%	-	22%
Large Cigar	7%	4%	6%	-
Dual/Poly Any Combustible Product Use	28%	40%	39%	43%
**As Recent** **(Greater than Past 6 months)**				
Cigarette	-	51%	43%	56%
LCC-T	47%	-	58%	62%
LCC-B	28%	41%	-	32%
Large Cigar	25%	30%	22%	-
Dual/Poly Any Combustible Product Use	49%	54%	50%	56%
**6 months or More, n**	379	341	243	166

## Data Availability

The data presented in this study are available upon reasonable request from the corresponding author.

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
