# Peer review of "Tobacco Whack-A-Mole: A Consumption Taxonomy of Cigar & Other Combustible Tobacco Products among a Nationally Representative Sample of Young Adults"

_ijerph, 2022, doi:10.3390/ijerph192215248_

Round 1

Reviewer 1 Report

This is an exceptional paper. It's extremely well written and carefully researched. It could be published as is, with one font issue noted below. I also have a comment for your consideration. 

It does not mention premium cigars. I assume that references to large cigars technically includes premium cigars, but given the age bracket of concern, it's likely a fraction of the large cigars measured. That said, if you think this captures premium cigars in a significant way, it should say so.

Even if this study barely captures premium cigars, the proposed change in measuring cigar use may have an additional implication for regulatory science.  The FDA's proposed warnings regarding premium cigars were struck down in court because the FDA failed to project the impact of the warnings, and a prior court decision struck down FDA regulations of premium cigars based on an absence of studies specific to premium cigars. If large cigar use should be measured through a broader window of time, that would change the way premium cigar use is measured in a significant way since many premium cigar users follow different usage patterns than cigarette users. I'm not certain this comment should affect your paper. I raise it for your consideration. 

In figure 1, the font for large cigars is, well, large, compared to the others.

Author Response

Point #1. It does not mention premium cigars. I assume that references to large cigars technically includes premium cigars, but given the age bracket of concern, it's likely a fraction of the large cigars measured. That said, if you think this captures premium cigars in a significant way, it should say so.

Response #1. We thank the reviewer for their helpful comments about premium cigars and the point of consideration. The reviewer is correct in that our large cigar category used premium brand names, such as Cohiba,  Macanudo, and Arturo Fuente. However, we referred to the category as large cigars because we did not specifically include the term “premium cigar” in our measure. Additionally, we are cognizant that the term large cigars may consist of both premium and non-premium brands. To clarify for our readers, we have added the following sentences to the manuscript:

“Our large cigar measure used premium cigar brand names, such as Cohiba, Macanudo, and Arturo Fuente, to assist respondents with the product category identification. Although premium cigar brands were used as examples, the use of non-premium large cigars also may have been captured in our measure.

Point #2. Even if this study barely captures premium cigars, the proposed change in measuring cigar use may have an additional implication for regulatory science.  The FDA's proposed warnings regarding premium cigars were struck down in court because the FDA failed to project the impact of the warnings, and a prior court decision struck down FDA regulations of premium cigars based on an absence of studies specific to premium cigars. If large cigar use should be measured through a broader window of time, that would change the way premium cigar use is measured in a significant way since many premium cigar users follow different usage patterns than cigarette users. I'm not certain this comment should affect your paper. I raise it for your consideration. 

Response #2. We thank the reviewer for this insightful comment and agree with it. We have added the following sentences to the “Implications for tobacco regulatory science” section:

Our findings suggest that the category of cigars, which includes cigarillos used with tobacco, cigarillos used with marijuana as blunts, and large cigars, including non-premium and premium brands, should be measured through a broader window of time because their usage patterns differ from other tobacco products (i.e., cigarettes). The suggested change in measuring cigar use may impact the evaluation of existing or future tobacco regulatory policy decisions (e.g., advertising restrictions) – thus assessing the impact of the policy on usage rates over an extended period (i.e., 3 months vs. past 30-days).

Point #3. In figure 1, the font for large cigars is, well, large, compared to the others.

Response #3. We thank the reviewer for pointing this out. The issue has been corrected in the figure.

Reviewer 2 Report

This is an innovative and important paper highlighting the need to improve measures of cigar use so that users are not left out of standard measures, thus reducing prevalence estimates, and having significant consequences for regulation. Thank you for doing this work and bringing this issue into the literature. I have one major comment and a few smaller ones.

1. The paper needs more connection to the idea that people who report using LCCs in the past 3 or 6 months is indicative of continued use rather than discontinued. How do we know (or might predict) that it's predictive of continued use, rather than categorized as discontinued use/quitting? My assumption is that this is connected to Table 3 results and using other products and the timing of their use, but this should be more explicitly highlighted throughout the intro, methods, results, discussion.

2. In the introduction, p. 2, line 56, "every day" or "someday" is not the same as past 30 day use. Those are 2 different 'current' use definitions. Just a minor revision to the sentence would be sufficient to make this accurate. The point made by the authors holds regardless.

3. Table 3 is very informative, but it is very dense and takes some time to orient to. Part of the issue seems to be that the equal/less than signs might be incorrect in some places. Additionally, there are some sub-sections that include both "greater than or equal to" and "less than or equal to", which results in overlap in time frame. Additionally, some times "more recent" is classified as "less than 3 months" whereas other times, it's "less than 6 months" and others "past 30 days". This makes the table confusing.

4. Please provide measurement information for demographics (e.g., gender identity, race, etc.)

Author Response

Point #1. The paper needs more connection to the idea that people who report using LCCs in the past 3 or 6 months is indicative of continued use rather than discontinued. How do we know (or might predict) that it's predictive of continued use, rather than categorized as discontinued use/quitting? My assumption is that this is connected to Table 3 results and using other products and the timing of their use, but this should be more explicitly highlighted throughout the intro, methods, results, discussion.

Response #1. We appreciate the reviewer’s comment. In the introduction, we provide evidence of the situational factors of LCC use, including social use and price, often associated with LCC use and how these may contribute to episodic use over an extended period. We use the fluctuating status of situational factors as a justification for the need to consider using measures that capture extended periods of LCC use. In the discussion section, we discuss how our consumption taxonomy framework highlighted the fluid nature of dual- and multiple-tobacco product use, as portrayed in Table 3.

While we appreciate the reviewer’s suggestion, it is beyond the scope of our study to say that extended use of LCCs (i.e., past 3 or 6 months) is predictive of continued product use. However, we acknowledge that our findings highlight a need to understand further the association between time since last use and continued product use. We also call for future studies to examine the association among these variables and situational factors, such as price and social use of the products.

Point #2. In the introduction, p. 2, line 56, "every day" or "someday" is not the same as past 30 day use. Those are 2 different 'current' use definitions. Just a minor revision to the sentence would be sufficient to make this accurate. The point made by the authors holds regardless.

Response #2. We thank the reviewer for this suggestion. The sentence has been edited and corrected.

Point #3. Table 3 is very informative, but it is very dense and takes some time to orient to. Part of the issue seems to be that the equal/less than signs might be incorrect in some places. Additionally, there are some sub-sections that include both "greater than or equal to" and "less than or equal to", which results in overlap in time frame. Additionally, sometimes "more recent" is classified as "less than 3 months" whereas other times, it's "less than 6 months," and others "past 30 days". This makes the table confusing.

Response #3. We thank the reviewer for this comment. We have updated the table as well as the text to add clarity.

Point #4. Please provide measurement information for demographics (e.g., gender identity, race, etc.)

Response #4. We thank the reviewer for this comment. We provided a description of sociodemographic characteristics in section 2.5.1.